# Evaluation of the effects of indocyanine green administration dose and speed on blood concentration and fluorescence in thoracoscopic segmentectomy

**Noriyuki Misaki** [ID]*, **Kayoko Sato, Kaede Yamada, Naoya Yokota, Natsumi Matsuura, Toshiki Yajima**

General Thoracic Surgery, Faculty of Medicine, Kagawa University, Kagawa, Japan

* gazyucup@yahoo.co.jp

## Abstract

Intravenous administration of indocyanine green (ICG) is commonly used for visualizing lungs along the intersegmental plane; however, its effectiveness can be limited, as poor fluorescence sometimes leads to suboptimal structural delineation. We aimed to evaluate the effects of ICG dose and injection speed on blood concentration and fluorescence intensity during thoracoscopic segmentectomy. This was a two-stage, single-center study involving 30 patients who underwent segmentectomy. In the first stage, the administration rate was fixed at 300 mL/h and the dose was investigated at 0.1 mg/kg (group L) and 0.2 mg/kg (group H). In the second stage, a rate of 900 mL/h (group F) was added to the dose determined in the first stage and the relationship between the dose, administration rate, and fluorescence was investigated. During the first stage, the blood concentration of ICG at the end of the marking was higher in group H; hence, the dose in group F was set to 0.2 mg/kg. In the subsequent analysis, evaluation was performed by comparing the three groups. The maximum blood concentration of ICG was significantly higher in group F (9.5 mg/L, p < 0.01) than in the other groups. There was no difference in the fluorescence intensity between the groups. The fluorescence intensity of group L at the end of the marking was low (123.4, p = 0.03). The intersegment plane was confirmed in all cases, and there was no difference in the marking rate between the groups. Although injection speed affected blood ICG concentration, it did not significantly enhance fluorescence intensity. The administered dose of ICG determined the duration of fluorescence. Results suggest a minimum ICG dose is required to sustain fluorescence, while excessively rapid injections may cause early threshold attainment. Clinically, maintaining blood ICG concentration within the 5–10 mg/L is best achieved with a relatively slow infusion rate of 300 mL/h. **Clinical trial registry number:** UMIN-CTR No UMIN000047255.

**Data availability statement:** All datasets are available from the Figshare database, DOI: https://doi.org/10.6084/m9.figshare.30738818.

**Funding:** This work was supported by JSPS KAKENHI [Grant Number: JP 22K08976]. The funder had no role in the study design, data collection and analysis, decision to publish, or preparation of the manuscript.

**Competing interests:** The authors have declared that no competing interests exist.

## Introduction

In recent multicenter clinical trials focusing on peripheral small-sized lung cancers, segmentectomy has been demonstrated to provide oncologic outcomes equivalent to or better than those of lobectomy, while preserving pulmonary function to a greater extent [1–3]. As a result, segmentectomy is increasingly being recognized as a standard surgical procedure for early-stage non-small cell lung cancer (NSCLC). Therefore, precise identification of the intersegmental plane is a critical factor in ensuring both oncologic safety and functional preservation [4].

However, since the intersegmental boundaries are generally not visible on the lung surface, various intraoperative techniques have been developed to delineate them. One widely adopted method involves the intravenous administration of indocyanine green (ICG) followed by near-infrared thoracoscopic imaging to visualize the perfused and non-perfused segments of the lung [4–7]. This technique enables real-time visualization of the intersegmental plane without the need for lung inflation and has been reported to achieve high identification rates (88–95%) along with favorable perioperative outcomes [8].

Despite its effectiveness, ICG often yields inconsistent visualization, with variable fluorescence duration and partial tissue perfusion, hindering reliable delineation of the intersegmental plane. Although patient-specific factors, including pleural adhesions, emphysematous changes, and interstitial lung disease, are factors known to contribute to the inconsistent outcomes of ICG [9,10], technical factors affecting the blood concentration of ICG, particularly dose and injection speed, are also believed to play a role.

We previously reported that constant-rate infusion using a syringe pump may facilitate the stabilization of ICG blood concentration and improve fluorescence quality [11]. In general, bolus injection speed varies considerably with the operator administering the injection, the type of syringe used, the injection route, and the resistance of the infusion line, leading to differences in the rise and peak of blood ICG concentration. Nevertheless, no systematic evaluation has been conducted to elucidate the relationship between ICG administration methods, resultant blood concentrations, and fluorescence intensity.

In this study, we initially performed an in vitro analysis to investigate the relationship between ICG blood concentration and fluorescence intensity. Based on these findings, we conducted a stepwise in vivo analysis comparing different ICG doses and injection speeds. The objective of this study was to clarify how these administration parameters influence blood concentration and fluorescence intensity during thoracoscopic segmentectomy.

## Materials and methods

### In vitro investigation of the relationship between the blood concentration of ICG and fluorescence intensity

We first conducted an in vitro study to evaluate the relationship between ICG blood concentration and fluorescence intensity. A stock solution was prepared by dissolving 25 mg of ICG dry powder (Daiichi-Sankyo Co., Tokyo, Japan) in 10 mL of distilled

water to obtain a concentration of 2500 mg/L. This solution was serially diluted and mixed with heparinized human whole blood to create 15 different concentrations ranging from 0.05 to 1250 mg/L. Each sample (0.2 mL) was dispensed into a well of a 96-well microplate (Greiner Bio-One International GmbH, Kremsmünster, Austria). Fluorescence was observed using an infrared thoracoscopic system (Visera Elite II, Olympus Co., Tokyo, Japan) at distances of 5 and 10 cm. Each measurement was performed in triplicate, and the median fluorescence value was calculated. The resulting curve showing the relationship between ICG concentration and fluorescence intensity served as the basis for selecting the dose range for the subsequent *in vivo* study.

### Image evaluation of fluorescence intensity

Image analysis followed a previously described method [11]. Intraoperative videos were converted to bitmap format using VideoStudio X10 (Corel Co., Ottawa, Canada). Still images were split into RGB channels using ImageJ 1.54p (National Institutes of Health, Maryland, USA), and fluorescence intensity was measured by selecting a region of interest in the green channel. Based on preliminary data [11], stable fluorescence visualization was defined as a fluorescence intensity exceeding 100 arbitrary units in perfused regions.

### Ethics

This study was approved by the ethics committee of Kagawa University (Approval No. 2020−165). The recruitment period for this study began on December 11, 2020 and ended on November 30, 2021. Written informed consent was obtained from all participants prior to their inclusion in the study. No minors were enrolled in this research. This was conducted in accordance with the principles of the Declaration of Helsinki, the International Conference on Harmonisation Good Clinical Practice guidelines, and all applicable laws and regulations. Personal data were de-identified to protect patient privacy.

### Patients

Eligible participants were adults aged 20 years or older who were scheduled to undergo thoracoscopic segmentectomy and provided written informed consent. Exclusion criteria included a history of allergy to ICG or iodine, liver dysfunction (Child–Pugh class B or higher), heart disease (New York Heart Association class III or higher), severe organ dysfunction, or determination by the treating physician that the patient was unsuitable for participation. The age threshold of 20 years was selected in accordance with local regulatory standards.

### Study design of stepwise comparison test

This was a single-center, exploratory study conducted using a stepwise comparison design. In Stage 1, the administration rate was fixed at 300 mL/h, and patients were alternately allocated to receive either 0.1 mg/kg (Group L) or 0.2 mg/kg (Group H) of ICG to evaluate dose-dependent effects. The results of blood concentration and fluorescence intensity were analyzed to determine the appropriate dose for Stage 2, in which a 0.2 mg/kg dose was adopted. In this stage, Group F received the same dose at a faster administration rate of 900 mL/h to assess the effect of injection speed. Each group included 10 patients. Patients were non-randomly allocated in an alternating manner within each stage. Because Stage 1 and Stage 2 were conducted sequentially, we also recognized the potential for temporal or secular-trend bias arising from changes over time in patient characteristics or clinical practice. However, the primary purpose of this study was exploratory dose and rate comparison. To avoid the risk of false-positive results due to multiple testing, the final analysis was conducted as a single, unified comparison across the three groups (Fig 1).

### Preoperative evaluation

All patients underwent preoperative respiratory function tests, electrocardiography, and blood tests. To evaluate metastasis, all patients received head magnetic resonance imaging and contrast-enhanced thoracoabdominal computed

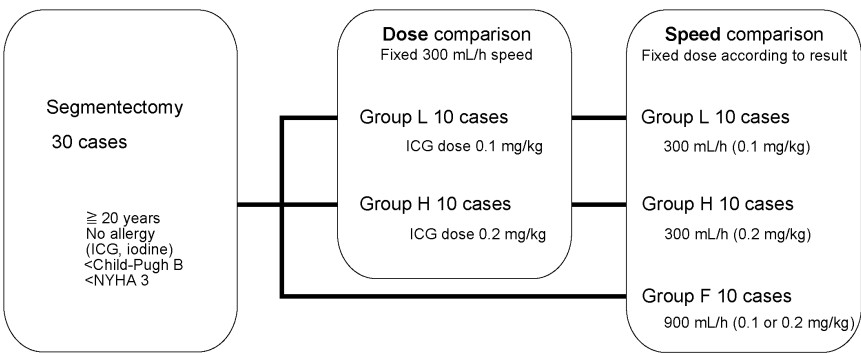

**Fig 1. Stepwise comparison test.**

tomography (CT). Additional fluorodeoxyglucose positron emission tomography was performed at the physician's discretion. CT images were reconstructed using a three-dimensional image analysis system (Synapse Vincent, Fujifilm Co., Tokyo, Japan) to create models of pulmonary vessels and bronchi. These models were used to identify the target pulmonary artery, vein, and segmental bronchus.

## ICG injection procedure and blood concentration monitoring

ICG was prepared at a concentration of 2.5 mg/mL by dissolving 25 mg of dry powder in 10 mL of distilled water. The solution was loaded into a 50-mL syringe. A second syringe with saline was prepared for flushing. Both were used with an infusion pump (Terufusion Syringe Pump Type 35; TE352, Terumo Co., Tokyo, Japan). The designated ICG dose was administered via a peripheral venous catheter placed in the forearm. After injection, the line was flushed with saline at the same rate.

Non-invasive monitoring of blood ICG concentration was performed using the DDG 3000 system (Nihon Kohden Co., Tokyo, Japan). This device measures changes in optical absorption at the nasal wing using two wavelengths (805 and 890 nm), based on the principle of pulse oximetry. Because indocyanine green exhibits peak absorption at 805 nm while the extinction coefficients of oxyhemoglobin and reduced hemoglobin are nearly equal at this wavelength, the ratio of ICG to hemoglobin concentration can be calculated for each pulse. Using the hemoglobin value obtained by conventional methods, absolute ICG concentrations can thus be estimated non-invasively and continuously [12]. After induction of general anesthesia, a sensor was placed on the nasal wing of each patient, and measurement was initiated simultaneously with the start of ICG injection. Continuous monitoring was performed for 15 minutes. The maximum blood concentration of ICG and the level at the end of fluorescence marking were recorded.

## Surgical procedure

All procedures were performed under general anesthesia with the patient in the lateral decubitus position. A three- or four-port thoracoscopic approach was employed: a 3–4 cm access port at the fourth intercostal space along the anterior axillary line, a 5-mm assistant port at the third intercostal space, a 2-cm assistant port at the sixth intercostal space, and a 2-cm camera port at the seventh intercostal space along the mid-axillary line. Infrared imaging was performed using the Visera Elite II system (Olympus Corp., Tokyo, Japan).

Pleural adhesions were dissected as necessary. The pulmonary artery supplying the target segment was ligated and divided. Division of the corresponding pulmonary vein and bronchus was not deemed mandatory. After completion of the pulmonary artery division, the lung was returned to its anatomical position, and ICG was administered. The

intersegmental plane was visualized under infrared thoracoscopy, and markings were placed using the soft coagulation mode of an electrosurgical unit. Successful marking was defined as the identification and marking of all targeted intersegmental planes.

## Statistical analyses

Data are presented as medians with ranges (interquartile ranges). Owing to the small sample size, non-parametric tests were used without assessing normality. Mann–Whitney U tests were applied for continuous variables in two-group comparisons, and Fisher's exact tests were used for categorical variables. Three-group comparisons were analyzed using the Kruskal–Wallis test (continuous variables) and Fisher's exact test (categorical variables). Bonferroni correction was applied for pairwise comparisons in the three-group analyses, and all reported p-values for pairwise tests represent Bonferroni-adjusted values. A post-hoc power analysis was performed with an alpha error of 0.05 and a power of 0.8, assuming a 1:1 sample size ratio, to estimate the sample size required for detecting differences in future studies. Statistical analysis was performed using R version 3.4.1 (R Foundation for Statistical Computing, Vienna, Austria) with the EZR package. A two-tailed p-value of ≤ 0.05 was considered statistically significant.

## Results

### In vitro fluorescence analysis

The fluorescence intensity increased with the concentration, peaking at 5 mg/L (5 cm) and 10 mg/L (10 cm), while decreasing at higher concentrations (Fig 2A). Fluorescence remained above 100 arbitrary units within the range of 1–40 mg/L (5 cm) and 5–20 mg/L (10 cm) (Fig 2B). Based on these results, the target blood ICG concentration in the clinical trial was set at 5–10 mg/L.

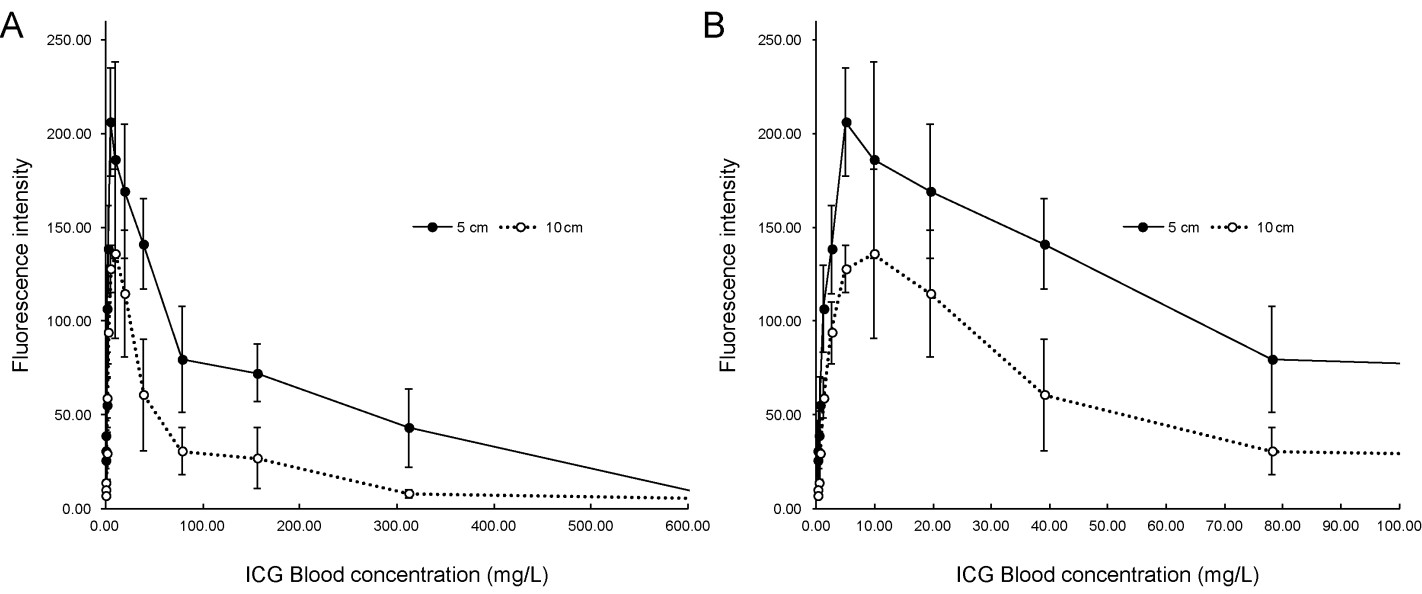

**Fig 2. Graph of the relationship between ICG concentration in the blood and fluorescence intensity. (A)** Luminance rose sharply to form a peak in the low concentration range and then declined gradually toward higher concentrations. **(B)** The peak is enlarged. A peak in the ICG concentration is visible at 5–10 mg/L. ICG, indocyanine green.

### In vivo analysis

The characteristics of the cases in each group are shown in Table 1. The distribution of lung lobes corresponding to the resected segments was balanced, with six upper-lobe cases and four lower-lobe cases.

### Stage 1: Dose comparison (Group L vs. Group H)

In the initial dose-comparison phase (fixed speed at 300 mL/h), 10 patients each were allocated to Group L (0.1 mg/kg) and Group H (0.2 mg/kg). There was no significant difference in maximum blood ICG concentration: 3.6 mg/L (3.2–5.7) vs. 5.7 mg/L (3.9–6.5), p = 1.0, and the ICG blood concentration at the end of the marking 1.2 mg/L (1.0–3.0) vs. 3.2 mg/L (2.2–3.6), p = 0.16. However, the fluorescence intensity at the end of marking was significantly lower in Group L than in Group H: 123.4 (110.7–145.2) vs. 197.8 (173.5–219.9), p = 0.03. Based on these findings, the dose for the next phase was fixed at 0.2 mg/kg. The p-values shown for pairwise comparisons are Bonferroni-adjusted.

### Stage 2: Rate comparison (Group F vs. Groups H and L)

In the second phase, Group F (0.2 mg/kg, 900 mL/h) was added. The maximum blood ICG concentration was significantly higher in Group F than in the other groups (Table 2): 9.5 mg/L (8.4–11.5) vs. 5.7 mg/L (Group H) and 3.6 mg/L (Group L), p < 0.001 (Fig 3A). However, the fluorescence intensity at maximum blood concentration did not differ significantly between the groups (p = 0.19) (Fig 3B). At the end of marking, the ICG concentrations were also comparable across all groups, with no statistically significant differences observed (Fig 4A). The fluorescence intensity at the end of marking in Group L 123.4 (110.7–145.2) was also significantly lower than that in Group H 197.8 (173.5–219.9) (p = 0.03) (Fig 4B).

A post-hoc analysis was conducted, and the sample size was examined based on the results. Because the administration speed was thought to affect the maximum blood concentration, the required sample size was calculated for groups F and H, which had different speeds but the same dose, and was found to be seven cases. On the other hand, since the dose was thought to affect the blood concentration at the end of marking, the required number of cases was calculated for group H and group L, which had the same speed but different doses, and was found to be 27 cases.

**Table 1. Patient characteristics and perioperative-related factors.**

| Group (cases) | F (n = 10) | H (n = 10) | L (n = 10) | p-value |
|---|---|---|---|---|
| Age, years | 72.5 (71-82) | 67.5 (61-72) | 73.0 (64-86) | 0.14 |
| Female | 5 | 5 | 5 | 1 |
| Height, cm | 158.0 (148.9-165.8) | 163.0 (156.1-169.1) | 160.0 (153.0-166.1) | 0.68 |
| Weight, kg | 55.0 (45.0-65.6) | 60.0 (53.8-78.0) | 58.0 (48.0-63.0) | 0.43 |
| Respiratory complication | 5 | 2 | 5 | 0.27 |
| Charlson Comorbidity Index | 6 (5–8) | 3.5 (3–5) | 5 (3–6) | 0.06 |
| Operated lung lobe Upper: Lower | 6:4 | 6:4 | 6:4 | 1.00 |
| Operation time, min | 198 (181.8-255.5) | 183 (129.0-224.0) | 192.5 (151.3-217.8) | 0.57 |
| Bleeding, mL | 50.0 (11.8-124.8) | 37.5 (3.5-58.3) | 25.0 (0-87.5) | 0.47 |
| Postoperative comorbidity | 2 (prolonged leakage) | 1 (prolonged leakage) | 1 (bronchial fistula) | 1 |
| Hospital stay, days | 5 (4-7) | 6 (4-13) | 6 (5-7) | 0.80 |

Group F: 900 mL/h speed and 0.2 mg/kg dose. Group H: 300 mL/h speed and 0.2 mg/kg dose. Group L: 300 mL/h speed and 0.1 mg/kg dose

**Table 2. Values related to ICG administration.**

| Group (cases) | F (n = 10) | H (n = 10) | L (n = 10) | p-value |
|---|---|---|---|---|
| **Maximum ICG blood concentration (mg/L)** | 9.5 (8.4-11.5) | 5.7 (3.9-6.5) | 3.6 (3.2-5.7) | <0.001 |
| **ICG blood concentration at the end of marking (mg/L)** | 3.1 (3.0–3.4) | 3.2 (2.2–3.6) | 1.2 (1.0–3.0) | 0.07 |
| **From IV to fluorescence (s)** | 24.0 (20.5-57.3) | 36.0 (27.5–54.5) | 62.0 (36.0–97.8) | 0.16 |
| **Taken for marking (min)** | 3.0 (1.6-3.2) | 1.8 (1.4–3.0) | 2.2 (1.5–2.5) | 0.55 |
| **Luminance at maximum blood concentration** | 152.2 (123.8-187.8) | 186.2 (159.7-218.0) | 174.7 (164.5-189.6) | 0.19 |
| **Luminance at the end of marking** | 164.0 (120.8–204.0) | 197.8 (173.5–219.9) | 123.4 (110.7-145.2) | 0.03 |
| **Success of all markings** | 5 | 7 | 8 | 0.35 |

ICG, indocyanine green; IV, intravenous.

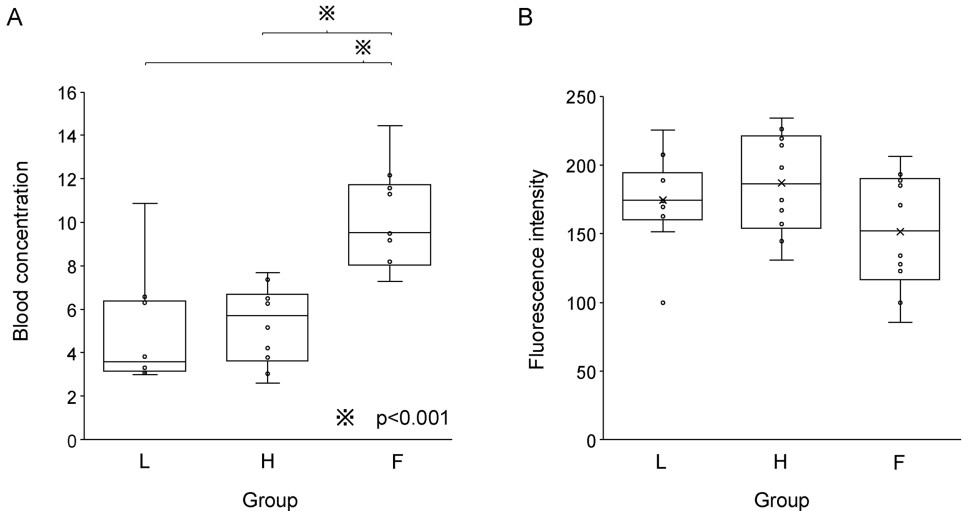

**Fig 3. Indocyanine green blood concentration and fluorescence intensity at maximum blood concentration. (A)** Group F had a higher blood concentration than the other two groups. **(B)** No difference in fluorescence intensity between the three groups.

Furthermore, the number of cases was calculated to be 24 when the fluorescence intensity at the time of the highest blood concentration was calculated for groups F and H, and 12 when the fluorescence intensity at the end of marking was calculated for groups H and L.

### Reference outcome: Marking success rate

Although not a primary outcome of this study, the success rate of intersegmental plane marking was recorded as a reference value. Marking was successful in 8/10 patients (80%) in Group L, 7/10 (70%) in Group H, and 5/10 (50%) in Group F. These differences were not significant (p = 0.35).

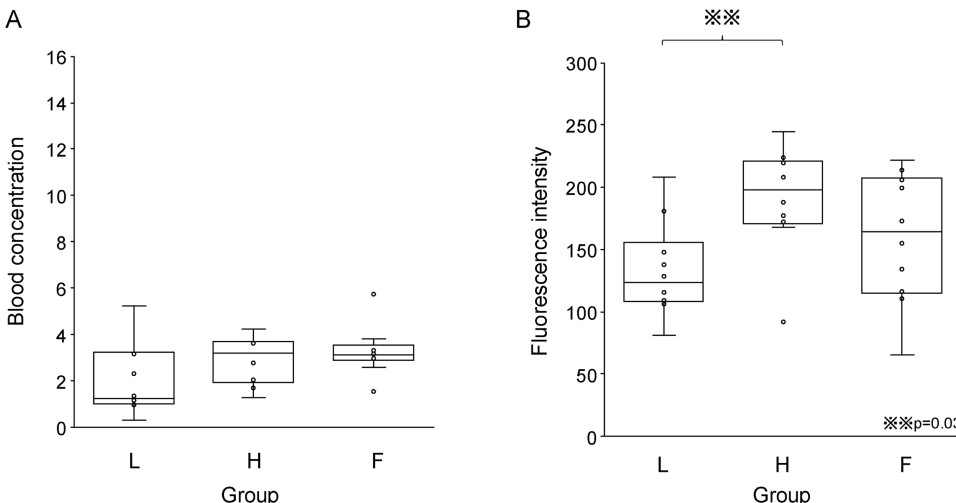

**Fig 4. Indocyanine green blood concentration and fluorescence intensity at the end of marking. (A)** No difference in blood concentration between the three groups. **(B)** Group L had lower fluorescence intensity than Group **H.**

## Discussion

Infrared video-assisted thoracoscopic surgery is a simple and widely used method for identifying the intersegmental plane after intravenous ICG [4–12]; however, the ideal dose and administration rate of ICG remain unclear. This study aimed to clarify how ICG blood concentration and fluorescence intensity were affected by the dose and injection rate during thoracoscopic segmentectomy. We examined the kinetics and optical behavior of ICG during intersegmental plane identification.

In vitro findings demonstrated that fluorescence intensity increased rapidly at low ICG concentrations, peaking at approximately 5 mg/L when observed from a distance of 5 cm and at 10 mg/L from 10 cm. Beyond these concentrations, the intensity gradually declined. These results suggest that concentrations of 5–10 mg/L provide optimal contrast for thoracoscopic visualization at typical observation distances (5–10 cm). Higher concentrations did not enhance visibility, indicating that low to moderate ICG concentrations are preferable for achieving sharp fluorescence contrast.

Fluorescence contrast observed in vitro helps explain the phenomena observed in vivo. In the low-dose group, ICG blood concentration tended to decrease, and fluorescence intensity appeared to decline over time. This may account for the occasionally observed rapid disappearance of fluorescence [6] and poor fluorescence visualization [7]. Fluorescence duration is shorter when ICG concentration fails to reach the optimal 5–10 mg/L range. Additionally, the injection speed also influenced the peak blood concentration of ICG. However, this increase in concentration did not lead to improved fluorescence intensity. ICG blood concentration above 5–10 mg/L does not further enhance fluorescence. Despite differences in maximum blood concentration, the fluorescence intensity plateaued, suggesting that a flow rate of approximately 300 mL/h is sufficient for fluorescence observations, while rapid administration may cause unnecessarily high blood concentrations. This may not improve fluorescence and could raise ICG blood concentration in the ischemic areas, causing bleeding across the intersegmental plane.

This study has several limitations. First, because fluorescence was evaluated using a single endoscopic system, the generalizability of our findings is impacted. Results may vary with different endoscopic systems and observational environments. Second, measured fluorescence intensity may vary by site and evaluation phase. Poor fluorescence visualization may be affected by factors such as emphysematous changes, carbon deposition, and impaired hepatic and cardiac functions. This study focused on ICG blood concentration and fluorescence intensity; therefore, we did not investigate

whether inadequate fluorescence and failure to identify the intersegmental plane were due to suboptimal concentrations or whether excessive concentrations caused fluorescence spillovers from adjacent segments, as this was beyond the scope of the present study. Our findings suggest that the blood concentration range for stable fluorescence is surprisingly narrow; neither high nor low concentrations alone ensure area identification. Nevertheless, ICG blood concentration levels and tissue fluorescence intensity significantly affect the area identification rate. Additionally, this study used a non-random allocation method and sequential enrollment across Stage 1 and Stage 2. Therefore, the possibility of temporal or secular-trend bias cannot be excluded, as changes over time in patient background, perioperative management, or clinical practice may have influenced the observed outcomes. Another limitation is that the resected segments varied among patients, and segment-specific differences in fluorescence cannot be excluded. However, when categorized by lung lobe, the distribution was balanced, with six upper-lobe cases and four lower-lobe cases, minimizing potential lobe-level bias. Finally, this study was conducted at a single center with a small number of cases, which constitutes a certain limitation. Therefore, the evaluation of the marking rate for all intersegmental planes should be considered as a reference rather than definitive data. Future studies with larger sample sizes are necessary to validate these findings.

Using the ICG-based intersegmental plane identification method, we assessed how dose and administration rate affected blood concentration. More distinct intersegmental observations may be possible by appropriately adjusting the dosage and speed. The intersegmental plane identification rate in daily clinical practice may be expected to improve by adopting a method of administration that stabilizes the blood concentration and avoids unnecessarily high concentrations. Maintaining blood ICG concentration within the 5–10 mg/L "sweet spot" for strong fluorescence is best achieved with a relatively slow infusion rate of 300 mL/h, which we recommend as a practical approach in clinical practice.

## Author contributions

**Conceptualization:** Noriyuki Misaki.

**Data curation:** Noriyuki Misaki.

**Formal analysis:** Noriyuki Misaki.

**Funding acquisition:** Noriyuki Misaki.

**Investigation:** Kayoko Sato, Kaede Yamada, Naoya Yokota.

**Methodology:** Noriyuki Misaki.

**Project administration:** Noriyuki Misaki, Natsumi Matsuura.

**Resources:** Noriyuki Misaki.

**Supervision:** Toshiki Yajima.

**Visualization:** Noriyuki Misaki.

**Writing – original draft:** Noriyuki Misaki.

**Writing – review & editing:** Noriyuki Misaki.

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
