## [Decision Letter · Decision Letter 0]

26 Nov 2025

Dear Dr. Misaki,

Thank you for submitting your manuscript to PLOS ONE. After careful consideration, we feel that it has merit but does not fully meet PLOS ONE’s publication criteria as it currently stands. Therefore, we invite you to submit a revised version of the manuscript that addresses the points raised during the review process.

We look forward to receiving your revised manuscript.

Kind regards,

Luca Bertolaccini, M.D., Ph.D.

Academic Editor

PLOS ONE

Journal Requirements:

This work was supported by JSPS KAKENHI [Grant Number: JP 22K08976].

This work was supported by JSPS KAKENHI Grant Number JP 22K08976.

This work was supported by JSPS KAKENHI [Grant Number: JP 22K08976].

Additional Editor Comments :

The reviewers have emphasised issues that require a careful and thorough manuscript revision.

No commitment to publication can be made at this point.

Reviewers' comments:

Reviewer's Responses to Questions

**Comments to the Author**

1. Is the manuscript technically sound, and do the data support the conclusions?

Reviewer #1: Yes

Reviewer #2: Yes

2. Has the statistical analysis been performed appropriately and rigorously?

Reviewer #1: Yes

Reviewer #2: No

3. Have the authors made all data underlying the findings in their manuscript fully available?

Reviewer #1: No

Reviewer #2: No

4. Is the manuscript presented in an intelligible fashion and written in standard English?

Reviewer #1: Yes

Reviewer #2: No

Reviewer #1: I want to start by acknowledging that you have tackled a very practical question in segmentectomy. The overall experimental setup is clinically relevant. Your focus on quantitative ICG behavior is refreshing and useful. I’m sharing the comments below in a constructive spirit:

1. You already present the 5–10 mg/L “sweet spot” and the adequacy of 300 mL/h quite clearly. It may help readers if you end the Abstract and the Discussion with one practical recommendation (something like a clear “this is what we would do clinically based on our findings”).

2. You explain the stepwise design, but it would help to spell out just a bit more transparently how Stage 1 influenced Stage 2. Was patient accrual sequential? And what exactly triggered the choice of conditions in Stage 2?

3. For Stage 1 you say randomization was used, but there’s no detail. Could you clarify how randomization/allocation was actually implemented? Also, for Stage 2/Group F, was every subsequent patient assigned to that group?

4. The in vitro and in vivo parts are both strong, but I think one short sentence connecting concentrations for each group (L/H/F) to the 5–10 mg/L window would make your overall message feel more cohesive.

5. The marking success rate didn’t differ significantly between groups, so the Discussion statement about ICG concentration “significantly affecting” area identification probably needs softening.

6. Several findings hover near significance (e.g., P=0.07) but are described a bit too confidently. I’d suggest framing them more as exploratory/suggestive trends.

7. Since you rely heavily on the DDG 3000 system, consider providing a brief note about its validation/calibration (with a reference if available).

8. Some mechanistic explanations (mixing, bleeding risk, central vs peripheral injection) read a little stronger than what your data directly evaluate. You may just soften the tone or label them as hypotheses.

9. You mention doing a post-hoc power analysis but don’t report the results. Consider providing a quick sentence summarizing it. Also, please make clear where Bonferroni adjustments are applied.

Thank you for sharing your work with me. I genuinely enjoyed reading it.

Reviewer #2: Review of the paper "Evaluation of the effects of indocyanine green administration dose and speed on blood concentration and fluorescence in thoracoscopic segmentectomy". The authors presented a study with two phases: the first in vitro and the second in vivo evaluating the best dose and speed of ICG for identifying the intersegmental plane during segmentectomy. The authors found that the dose of 0.2 mg/kg administered at high speed is associated with a higher concentration of blood ICG and a better visualization of the intersegmental plane. In my opinion, the study is good, but with limited interest because the ICG dose is quite standardized, effective at high dose and without complications or patients risk because the safety profile of the drug is high.

I suggest clarifying some aspects:

1) how was the number of patient selected?

2) I suggest including the type of segmentectomies perfomed

3) I suggest using the interquartile range associated with the median

(what does this mean? ). If published, this will include your full peer review and any attached files.). If published, this will include your full peer review and any attached files.

**Do you want your identity to be public for this peer review?** For information about this choice, including consent withdrawal, please see our For information about this choice, including consent withdrawal, please see our Privacy Policy .

Reviewer #1: **Yes:** Savvas LampridisSavvas Lampridis

Reviewer #2: **Yes:** Stefano BongiolattiStefano Bongiolatti

---

## [Decision Letter · Decision Letter 1]

15 Feb 2026

Dear Dr. Misaki,

Thank you for submitting your manuscript to PLOS ONE. After careful consideration, we feel that it has merit but does not fully meet PLOS ONE’s publication criteria as it currently stands. Therefore, we invite you to submit a revised version of the manuscript that addresses the points raised during the review process.

We look forward to receiving your revised manuscript.

Kind regards,

Luca Bertolaccini, M.D., Ph.D.

Academic Editor

PLOS One

Journal Requirements:

Additional Editor Comments:

I would urge the authors to carefully read the remarks of Reviewers and consider these in the revised paper.

Reviewers' comments:

Reviewer's Responses to Questions

**Comments to the Author**

Reviewer #1: (No Response)

Reviewer #2: All comments have been addressed

2. Is the manuscript technically sound, and do the data support the conclusions?

Reviewer #1: Yes

Reviewer #2: Yes

3. Has the statistical analysis been performed appropriately and rigorously?

Reviewer #1: Yes

Reviewer #2: Yes

4. Have the authors made all data underlying the findings in their manuscript fully available?

Reviewer #1: Yes

Reviewer #2: Yes

5. Is the manuscript presented in an intelligible fashion and written in standard English?

Reviewer #1: Yes

Reviewer #2: Yes

Reviewer #1: Thank you for the revisions and responses. Your manuscript is now much improved, and most of the major issues have been addressed. However, there are a few points that still need your attention:

1. You have correctly clarified that patients were allocated alternately rather than randomly, and I appreciate the correction of the earlier wording. However, I believe this point still needs to be handled more clearly in the manuscript itself. Please describe this as non-random allocation and add a brief acknowledgment of the potential for temporal or secular-trend bias, given the sequential nature of Stage 2 following Stage 1.

2. Although you now state that Bonferroni correction was applied for pairwise comparisons, it still seems unclear to me whether the reported “significant” p-values are adjusted or unadjusted. For example, p-values around 0.03 would not remain significant after Bonferroni correction for three pairwise tests. Please clarify this. Either report Bonferroni-adjusted p-values for pairwise comparisons or distinguish between overall test significance and exploratory/unadjusted pairwise findings.

3. I think there is an inconsistency between the rebuttal and the manuscript regarding how data are summarised. In some places you state that values are presented as median (IQR), while elsewhere the Methods describe medians with min–max. Please standardise this.

4. Your response explaining the distribution of upper versus lower segmentectomies is reasonable, but I think this information should be summarised in the manuscript itself rather than in the response letter or external dataset only (even one sentence in the baseline characteristics or limitations).

Many thanks.

Reviewer #2: This version of the paper has been effectively improved and I do not have any further comment. However, a list of performed segments could be informative for the reader, as I previously suggested.

(what does this mean? ). If published, this will include your full peer review and any attached files.). If published, this will include your full peer review and any attached files.

**Do you want your identity to be public for this peer review?** For information about this choice, including consent withdrawal, please see our For information about this choice, including consent withdrawal, please see our Privacy Policy .

Reviewer #1: **Yes:** Savvas LampridisSavvas Lampridis

Reviewer #2: **Yes:** Stefano BongiolattiStefano Bongiolatti

---

## [Author Response · Author response to Decision Letter 2]

11 Mar 2026

Reviewer #1

1. You have correctly clarified that patients were allocated alternately rather than randomly, and I appreciate the correction of the earlier wording. However, I believe this point still needs to be handled more clearly in the manuscript itself. Please describe this as non-random allocation and add a brief acknowledgment of the potential for temporal or secular-trend bias, given the sequential nature of Stage 2 following Stage 1.

Answer

We appreciate the reviewer’s helpful suggestion. In accordance with your comment, we have revised the Methods section to explicitly state that patients were non-randomly allocated in an alternating manner within each stage. We also added a brief explanation acknowledging the potential for temporal or secular‑trend bias due to the sequential implementation of Stage 1 followed by Stage 2.

Specifically, we added the following sentence to the Methods section:

“Patients were non-randomly allocated in an alternating manner within each stage. Because Stage 1 and Stage 2 were conducted sequentially, we also recognized the potential for temporal or secular‑trend bias arising from changes over time in patient characteristics or clinical practice. However, the primary purpose of this design was exploratory dose and rate comparison.”

In addition, we incorporated the following statement into the Limitations section of the 　　Discussion to further clarify the potential impact on interpretation:

“This study used a non-random allocation method and sequential enrollment across Stage 1 and Stage 2. Therefore, the possibility of temporal or secular‑trend bias cannot be excluded, as changes over time in patient background, perioperative management, or clinical practice may have influenced the observed outcomes.”

We believe these revisions address the reviewer’s concern and improve the clarity and transparency of the study design.

2. Although you now state that Bonferroni correction was applied for pairwise comparisons, it still seems unclear to me whether the reported “significant” p-values are adjusted or unadjusted. For example, p-values around 0.03 would not remain significant after Bonferroni correction for three pairwise tests. Please clarify this. Either report Bonferroni-adjusted p-values for pairwise comparisons or distinguish between overall test significance and exploratory/unadjusted pairwise findings.

Answer

Thank you for pointing this out. We apologize for the lack of clarity in the previous version. All p‑values reported for the pairwise comparisons in the three‑group analysis are Bonferroni-adjusted values calculated directly by the statistical software. Therefore, the p‑values around 0.03 shown in the Results section represent adjusted p‑values and remain statistically significant after correction. To avoid misunderstanding, we have revised the Methods and Results sections to explicitly state that Bonferroni‑adjusted p‑values are reported for all pairwise comparisons. In response to your suggestion, we re‑examined all statistical procedures, including the pairwise comparisons and the evaluation of blood concentration and fluorescence intensity. We confirmed that the statistical software outputs Bonferroni‑adjusted p‑values for the pairwise tests, and we have corrected the manuscript to clearly indicate that all reported p‑values are adjusted values.

3. I think there is an inconsistency between the rebuttal and the manuscript regarding how data are summarised. In some places you state that values are presented as median (IQR), while elsewhere the Methods describe medians with min–max. Please standardise this.

Answer

Thank you for pointing out this inconsistency. In response to your comment, we carefully reviewed the manuscript and standardised all summary statistics to be presented as median (IQR). We revised both the Methods section and all relevant tables and text to ensure consistency throughout the manuscript.

We appreciate your attention to detail, which has helped improve the clarity and uniformity of our data presentation.

4. Your response explaining the distribution of upper versus lower segmentectomies is reasonable, but I think this information should be summarised in the manuscript itself rather than in the response letter or external dataset only (even one sentence in the baseline characteristics or limitations).

Answer

Thank you for this helpful suggestion. In response to your comment, we have incorporated this information directly into the manuscript. Specifically, we added the distribution of upper‑ versus lower‑lobe segmentectomies to the baseline characteristics table to ensure that readers can easily confirm the balance between groups.

In addition, we added a brief statement to the Limitations section noting that although the resected segments varied, the corresponding lung lobes were evenly distributed (six upper‑lobe and four lower‑lobe cases), thereby reducing the likelihood of lobe‑level anatomical bias.

We believe these additions improve the transparency and completeness of the manuscript.

---

## [Editor Report · Decision Letter 2]

12 Mar 2026

Evaluation of the effects of indocyanine green administration dose and speed on blood concentration and fluorescence in thoracoscopic segmentectomy

PONE-D-25-54320R2

Dear Dr. Misaki,

We’re pleased to inform you that your manuscript has been judged scientifically suitable for publication and will be formally accepted for publication once it meets all outstanding technical requirements.

Kind regards,

Luca Bertolaccini, M.D., Ph.D.

Academic Editor

PLOS One
---

## [Editor Report · Acceptance letter]

PONE-D-25-54320R2

PLOS One

Dear Dr. Misaki,

I'm pleased to inform you that your manuscript has been deemed suitable for publication in PLOS One. Congratulations! Your manuscript is now being handed over to our production team.

Kind regards,

on behalf of

Dr. Luca Bertolaccini

Academic Editor

PLOS One